# Hepatic Gene Expression of Angiogenic and Regeneration Markers in Cats with Congenital Portosystemic Shunts (CPSS)

**DOI:** 10.3390/vetsci11030100

**Published:** 2024-02-26

**Authors:** Michael S. Tivers, Samantha M. Mirczuk, Abigail Charlesworth, Lauren Wood, Emi N. Barker, Victoria J. Lipscomb, Robert C. Fowkes

**Affiliations:** 1Paragon Veterinary Referrals, Paragon Business Village, Red Hall Cres, Wakefield WF1 2DF, UK; mickey.tivers@paragonreferrals.co.uk; 2Langford Vets, University of Bristol, Langford BS40 5DU, UK; emi.barker@bristol.ac.uk; 3Endocrine Signalling Group, Comparative Biomedical Sciences, Royal Veterinary College, University of London, Royal College Street, London NW1 0TU, UK; samantha.byers@admin.cam.ac.uk (S.M.M.);; 4Bristol Vet School, University of Bristol, Langford BS40 5DU, UK; 5Clinical Sciences & Services, Hawkshead Lane, North Mymms, Hatfield, Hertfordshire AL10 8TY, UK; vlipscomb@rvc.ac.uk; 6Small Animal Clinical Sciences, College of Veterinary Medicine, Michigan State University, 736 Wilson Road, East Lansing, MI 48824, USA

**Keywords:** hepatic encephalopathy, liver disease, gene expression

## Abstract

**Simple Summary:**

Cats can be born with a rare vascular malformation, whereby an abnormal vessel diverts blood from the gut around the liver (congenital portosystemic shunt), preventing detoxification and leading to numerous clinical signs. Surgical approaches are usually successful in closing the shunting vessel and improving liver function in affected cats. Although this disorder is well characterised in dogs, the mechanisms involved in altered liver function in cats with these abnormalities is poorly understood. The objective of this study was to evaluate whether the expression of genes associated with the development and function of the liver is different in the liver of cats with portosystemic shunts. Results of liver biopsy samples from eighteen cats with portosystemic shunts were compared to those of ten cats without portosystemic shunts. Increased expression of five genes associated with liver regeneration, angiogenesis, metabolism, and urea cycle regulation was observed in cats with portosystemic shunts. These findings are in contrast to the changes documented in dogs with portosystemic shunts, and suggests that liver regeneration, angiogenesis, and urea metabolism may be controlled by different mechanisms in cats.

**Abstract:**

Congenital portosystemic shunts (CPSS) are vascular anomalies resulting in liver hypoplasia and hepatic insufficiency. Cats with CPSS typically show signs of hepatic encephalopathy associated with increased ammonia, inflammatory cytokines, and oxidative stress. Surgical attenuation of the CPSS results in improved liver function, resolution of clinical signs, and increased portal blood flow. Hepatic gene expression has not previously been investigated in cats with CPSS. Here, we compared the hepatic expression of genes involved in the urea cycle (*CPS1*, *NAGS*), angiogenesis (*VEGFR2*, *NPPA*, *NPR1*, *NPPC*, *NPR2*, *HIF1a*), liver regeneration (*SERPINB1*, *HGF*, *TGFβ*), and metabolism (*FGF21*) from a small series of cats (*n* = 18) with CPSS to that of control cats (*n* = 10). The expression of *TGFβ*, *VEGFR2*, *HGF*, *FGF21*, and *CPS1* was significantly elevated in liver biopsies from cats with CPSS. Cats that could only tolerate partial closure of their CPSS had increased hepatic expression of *SERPINB1*, *HIF1a*, and *NPR2* compared with those that could tolerate complete ligation. Furthermore, there were no significant correlations between gene expression and pre-operative plasma ammonia concentrations in cats with CPSS. The changes in hepatic gene expression in cats with CPSS are in direct contrast to those seen in dogs with CPSS, suggesting alternative mechanisms may be involved in mediating hepatic changes in cats with CPSS.

## 1. Introduction

Congenital portosystemic shunts (CPSS) are vascular anomalies that allow portal blood to bypass the liver [1,2,3]. Affected cats have liver hypoplasia and clinical signs associated with hepatic insufficiency including hepatic encephalopathy (HE) [4]. One of the major factors associated with HE is blood ammonia concentration, and this is typically increased in cats with CPSS [5,6]. However, the degree of hyperammonemia in dogs does not directly correlate with the severity of HE [7]. Surgical attenuation of the shunt is recommended in most cats to improve or restore normal portal blood flow to the liver [8,9,10]. Surgery typically results in an improvement in liver function, as assessed on clinicopathological tests including dynamic bile acids. This improvement in liver function is associated with the resolution of clinical sings. An improvement in portal blood flow has been demonstrated on intra-operative imaging in cats treated with partial ligation of their shunt [11]. 

In dogs with CPSS, successful surgery results in an increase in liver volume as assessed on computed tomography or magnetic resonance imaging [12,13]. This increased volume is assumed to be a result of liver regeneration. Several studies have measured the hepatic expression of markers of liver regeneration and angiogenesis in dogs with CPSS, providing evidence to support the role of liver regeneration in the response to surgery in this species [14,15,16,17,18,19,20]. Whilst similar processes are assumed to be involved in the response to surgery in cats with CPSS, this remains unproven. To the best of the authors’ knowledge, no studies have investigated hepatic gene expression in cats with CPSS.

Despite providing the potential to be curative, surgical treatment of feline CPSS can result in neurological complications distinct from HE (i.e., post-attenuation neurological signs, PANS) [21,22,23,24,25]. PANS was recorded in 62% of cats in one study [26], and are the main cause of post-operative mortality in cats with a rate of between 4.1% and 22.2%. Whilst another study reported that one third of cats treated surgically for CPSS developed seizures [11], long-term medical management with anticonvulsants may be required. To date, prognostic indicators of which cats may be more susceptible to developing PANS remain unknown. 

This was a pilot study to assess the feasibility of measuring the hepatic gene expression of various markers of liver regeneration and angiogenesis, and of urea cycle enzymes in a small cohort of cats with CPSS using a customised highly multiplexed RT-PCR (XP-PCR) assay to simultaneously detect multiple transcripts in individual samples. The aim of the study was to measure the mRNA expression of target genes in liver biopsies from cats with CPSS and to compare them with liver biopsy samples from control cats.

## 2. Materials and Methods

### 2.1. Patient Recruitment & Ethical Approval 

Cats diagnosed with a single CPSS were prospectively enrolled and underwent surgical attenuation of their CPSS at the Queen Mother Hospital for Animals or at the Langford Vets, Small Animal Referral Hospital between August 2007 and January 2018. All cats had their CPSS confirmed on ultrasound and/or computed tomography imaging prior to surgery. Control samples (two from surgical biopsies, eight at post-mortem (within 2 h) were obtained from cats that were undergoing liver biopsy for reasons unrelated to CPSS or from samples collected at post-mortem examination for inclusion in a biobank, stored in RNAlater™ (Sigma-Aldrich, Dorset, UK), held at Bristol Vet School from cats with a final diagnosis of a disease unrelated to CPSS. Cats had plasma ammonia measured at the discretion of their attending veterinary surgeon and pre-operative values were recorded, where available. Ammonia was measured using a Jenway 6310 Spectrophotometer (Bibby Scientific Limited, Staffordshire, UK). Samples were transported on ice to an onsite laboratory. The ammonia reference interval was 0–70 μmol/L. Ultrasound was used pre-operatively to diagnose a CPSS. The study was approved by the Royal Veterinary College (RVC) Ethics and Welfare Committee (URN 2013 1208). 

Exploratory coeliotomy and intra-operative mesenteric portovenography (IOMP) allowed confirmation and identification of, and dissection around, the CPSS. Portal pressure increase following temporary complete shunt occlusion was assessed. Complete ligation or partial attenuation was performed with the final portal pressure limited to the lowest of 8 mmHg above pro-occlusion value, double the pre-occlusion value, or 14 mmHg. Additionally, alterations in mean arterial pressure, central venous pressure, and heart rate, as well as intestinal or pancreatic congestion, were considered as evidence of portal hypertension. A wedge liver biopsy was taken from the peripheral edge of a liver lobe from each cat for routine histopathology. Surplus liver tissue was stored in RNAlater™ for subsequent analysis. In cats with continued shunting and/or clinical signs after surgery a revision surgery was offered to attempt complete shunt ligation once intrahepatic portal vasculature had developed further, typically around 3 months after the first surgery. A second liver biopsy was taken for routine diagnostic purposes at repeat surgery and surplus tissue was stored as above.

### 2.2. RNA Extraction and Multiplexed Reverse Transcription PCR (XP-PCR) 

Total RNA was extracted, quantified, and assessed for purity using an ND-100 spectrophotometer (Nanodrop, Thermo Fisher, Hemel Hempsted, UK) prior to being normalised to 100 ng/µL and prepared for multiplexed XP-PCR as previously described [27]. Specifically, target-specific reverse transcription and PCR amplification was performed as previously described and in accordance with manufacturer’s instructions (Beckman Coulter) [27], using oligonucleotide primers specific to hepatic genes (see Table 1). In brief, the reverse transcription reactions were prepared as detailed in the GenomeLab GeXP Start Kit (Beckman Coulter) and performed using a G-Storm thermal cycler, using the programme protocol: 48 °C for 1 min, 42 °C for 60 mins, and 95 °C for 5 min. From each reverse transcription reaction an aliquot was added to PCR master mix containing GenomeLab kit PCR master mix, Thermoscientific Thermo-Start *Taq* DNA polymerase. PCR reaction was performed using G-Storm GS1 thermal cycler with an activation step of 95 °C for 10 mins, followed by 35 cycles of 94 °C for 30 s, 55 °C for 30 s, and 70 °C for 60 s. Products were separated and quantified using CEQTM 8000 Genetic Analysis System, and GenomeLab Fragment Analysis software (eXpress Analysis Version 1.0.25, Beckman Coulter, High Wycombe, UK)). Absolute gene expression was converted to quantification relative to the housekeeping gene RSP7. 

### 2.3. Data Presentation & Analysis 

Analyses were performed using GraphPad Prism 10.0, with gene expression normalised to the housekeeping gene RPS7. Statistical comparisons were performed using Mann–Whitney *U* tests. Where available, simple linear regression analyses were performed on all relative gene expressions compared to pre-operative blood ammonia (μmol/L), using in-built equations within GraphPad Prism. For all tests, significance was set at a level of 5% (*P* ≤ 0.05). 

## 3. Results

### 3.1. Clinical Information

Liver samples were obtained from 18 cats with CPSS. Histopathological findings were consistent with portal vein hypoperfusion in all cats with CPSS, as previously described [28]. Four cats had a second sample taken at time of second surgery. A summary of clinical characteristics of cats enrolled in this study is found in Table 2. The median age of CPSS cats was 243 days (range 155 to 1309 days) and the median age of control cats was 2161 days (range 120 to 5353 days). Plasma ammonia was measured pre-operatively in 10 cats. The median plasma ammonia concentration was 277 µmol/L (range 134 to 564 µmol/L). 

### 3.2. Hepatic Gene Expression in Control Cats and CPSS Cats

Total RNA was successfully extracted from all liver samples as evidenced by detection of expressed housekeeper gene *RSP7*. Multiplex XP-PCR was performed using two custom-designed assays that allowed for simultaneous detection of multiple transcripts in a single sample. As shown (Figure 1A–N), expression of *HGF* (Figure 1A; 0.52 vs. 0.21, Mann–Whitney *U* test, *P* = 0.018)*, FGF21* (Figure 1C; 0.08 vs. 0.03, Mann–Whitney *U* test, *P* = 0.04)*, TGFβ* (Figure 1D; 0.45 vs. 0.06, Mann–Whitney *U* test, *P* = 0.005)*, VEGFR2* (Figure 1H; 0.74 vs. 0.51, Mann–Whitney *U* test, *P* = 0.003)*,* and *CPS1* (Figure 1J; 1.074 vs. 0.83, Mann–Whitney *U* test, *P* = 0.013) were all increased in liver biopsies from cats with CPSS compared to controls. In contrast, there were no significant differences in expression levels of *SERPINB1*, *Il1β*, *IL6*, *HIF1A*, *NAGS*, *NPPA*, *NPR1*, *NPR2*, or *NPPC*.

### 3.3. Comparison of Hepatic Gene Expression in CPSS Cats That Tolerated Complete or Partial Shunt Attenuation at First Surgery

Cats with CPSS were subdivided further into those that tolerated full shunt occlusion and those that could only tolerate partial shunt attenuation during surgery, to determine if the gene expression signatures were altered. Five cats tolerated full CPSS closure whereas thirteen cats tolerated only partial attenuation. Cats that could only tolerate partial shunt attenuation were found to have increased hepatic expression of *HIF1a* (0.7 vs. 0.92, Mann–Whitney *U* test, *P* = 0.048), *NPR2* (0.012 vs. 0.028, Mann–Whitney *U* test, *P* = 0.04), and *SERPINB1* (0.26 vs. 0.36, Mann–Whitney *U* test, *P* = 0.042). There were no other significant differences in hepatic gene expression observed between these two sub-cohorts of CPSS cats. 

### 3.4. Hepatic Gene Expression in Cats with CPSS Does Not Change Following Shunt Attenuation Surgery

Of the thirteen CPSS cats that received a partial attenuation at first surgery, liver biopsies were only available from four at second surgery. As shown (Figure 2), there were no significant differences in hepatic gene expression detected in these cats between first and second surgery.

### 3.5. Hepatic Gene Expression Cats with CPSS Does Not Correlate with Pre-Operative Ammonia Concentrations

We used a simple linear regression of gene expression against blood ammonia to determine whether hepatic gene expression in CPSS cats was related to pre-operative plasma ammonia concentrations. Hepatic gene expression levels did not correlate with pre-operative ammonia concentrations (Table 3).

## 4. Discussion

Congenital portosystemic shunts (CPSS) in cats can lead to life-threatening clinical signs and surgical intervention is recommended in most cats. Changes in hepatic gene expression have been detected in dogs with CPSS, and are related to abnormal angiogenesis, urea metabolism, inflammation, and liver regeneration [15,16,17,18,19,20]. Here, we measured the hepatic expression of 14 genes in cats with CPSS and compared them to hepatic gene expression in cats without CPSS. These included genes associated with liver regeneration (*HGF*, *TGFβ*), angiogenesis (*HIF1a*, *VEGFR2*), inflammation (*IL1β*, *IL6*, *SERPINB1*), urea cycle enzymes (*CPS1*, *NAGS*), natriuretic peptides (*NPPA*, *NPPC*, *NPR1*, *NPR2*), and metabolism (*FGF21*). Gene selection was based on those of interest from previous studies in dogs [14,16,17,18,19,20,29]. Initial optimization of the multiplex assay also included primers targeting *MAT2A*, although it was not possible to successfully multiplex this with other genes of interest in the assay. The study was unique in assessing the feasibility of measuring gene expression in liver samples from cats with CPSS. We were able to demonstrate that some components of the pathways associated with liver regeneration and angiogenesis are differentially expressed in cats with CPSS as compared to controls, with increased expression of *HGF*, *FGF21*, *TGFβ*, *VEGFR2*, and *CPS1*.

Liver regeneration is a very complex process, although hepatocyte growth factor (HGF) is a key component. There is growing evidence that the hepatic response to CPSS surgery in dogs occurs through liver regeneration. Studies have shown that *HGF* is downregulated in dogs with CPSS [16,18]. One study showed that the administration of recombinant canine HGF to dogs with CPSS was able to stimulate an increase in liver volume [30]. Another study found that hepatic expression of *HGF* increased following partial attenuation of the shunt and that serum HGF was significantly increased 24 h post-CPSS attenuation [16]. Successful shunt attenuation in cats is associated with similar clinical and clinicopathological improvements as seen in dogs [8]. It was assumed that the liver’s response to surgery would be similar in cats and dogs. The current study found that hepatic *HGF* expression was upregulated in cats with CPSS compared to controls, contrary to expectations. The possible reasons for this are unclear. Significant differences exist in how the liver of cats regulates hepatic processes such as glycolysis and gluconeogenesis compared to enzyme activity in the dog liver [31]. Pharmacokinetic studies show that various classes of drugs are metabolised differently in cats due to the absence of specific liver-associated enzymes [32]. The observed increase in liver expression of *HGF* in cats with CPSS might reflect an alternative mechanism of hepatic protection, or simply be a consequence of unrecognised differences in the disease process between species. Certainly, larger cohort studies are necessary to determine whether this increase in *HGF* expression is truly reflective of the hepatic response to CPSS in cats.

Studies in both dogs and cats have shown a significant improvement in intrahepatic portal vasculature as assessed by IOMP following partial ligation of CPSS [11,33]. It is hypothesised that this improvement occurs via angiogenesis, likely mediated by VEGF [14]. VEGF is a critical mediator of angiogenesis and acts predominantly through its receptor VEGFR2 [34]. A study found that using immunohistochemistry, endothelial VEGF was increased and VEGFR2 was decreased in dogs with CPSS compared with controls [35]. However, another study showed that in dogs with CPSS portal vasculature development was significantly positively associated with hepatic expression of *VEGFR2* [14]. Nevertheless, *VEGFR2* expression was not significantly different between dogs with CPSS and control dogs [14]. The current study found that that *VEGFR2* expression was increased in liver biopsies from cats with CPSS. Contrary to findings in dogs there was no association between whether cats with CPSS could tolerate complete attenuation of their shunt, but this might reflect the small numbers of cats in this pilot study. VEGFR2 expression has been used as a diagnostic marker in feline mammary carcinoma [36], where detection of circulating VEGF-A, VEGFR1, and VEGFR2 can be used as biomarkers of disease severity. Whether circulating levels of VEGF and/or VEGFR2 are altered in cats with CPSS remains to be seen, but this upregulation in liver biopsy samples could reflect an adaptive response to improve vascularization in hepatic tissue affected by CPSS.

We found a significant increase in hepatic expression of *FGF21* in cats with CPSS compared with control cats. This is a novel finding, and, to the authors’ knowledge, this marker has not been investigated in dogs with CPSS. *FGF21* expression in cats in general has not been extensively examined. Treatment of obese cats with FGF21 can reduce intrahepatic triglycerides and promote weight loss [37], but there are currently no data describing FGF21 in cats with CPSS. Human patients with liver disease exhibit an activation in the FGF21 pathway, potentially as a protective response [38], which suggests that circulating FGF21 levels might be useful as a biomarker for hepatic disease. The increase in *FGF21* expression in CPSS cats in our study could reflect an adaption, to prevent hepatotoxicity from endoplasmic reticulum (ER) stress [39], but further investigations of FGF21 function in cats is required to determine its potential as either a biomarker for liver disease or as a possible therapeutic agent.

Of the remaining hepatic genes that were significantly altered in liver biopsy samples from cats with CPSS, the increased expression of *CPS1* is particularly interesting. Elevated *CPS1* expression has been demonstrated in human patients with poor liver function and with hepatocellular injury [40,41]. However, dogs with CPSS expressed lower *CPS1* than unaffected control dogs, and this did not resolve upon successful shunt attenuation surgery [29]. CPS1 is a critical part of the urea cycle, involved in generating carbomyl phosphate from ammonia, bicarbonate, and ATP [42]. Mice with induced deficiency in hepatic CPS1 develop hyperammonemia within 4 weeks of induction and die [43]. Although arginine is an essential amino acid for both dogs and cats, cats have a higher requirement for arginine than dogs (and other species) because they maintain their urea cycle even during conditions of fasting or poor nutrition [44,45]. It is possible that the significantly enhanced expression of *CPS1* in cats with CPSS is reflective of this need to maintain the urea cycle, even under disease conditions. Our correlation studies of gene expression and pre-operative blood ammonia levels failed to show a significant relationship between ammonia concentrations and *CPS1* expression in our cats with CPSS. However, this may be a function of the small cohort size available (ammonia sampling was only performed in 10 cats).

We compared the hepatic gene expression of five cats that could tolerate complete closure of their CPSS with that of 13 cats that could only tolerate partial attenuation. The ability to tolerate complete CPSS closure can be considered a surrogate for good liver and portal vasculature development. Studies have shown that the hepatic expression of *VEGFR2* and *TLR4* were increased in dogs with CPSS that could tolerate complete closure compared with those that could only tolerate partial attenuation [14,15]. In our current study, we found the inability to tolerate complete CPSS closure was associated with the increased expression of *HIF1a*, *NPR2*, and *SERPINB1*. *SERPINB1* is an anti-inflammatory agent, released during time of stress, that acts to inhibit cellular damage from proteases [46]. Increased *SERPINB1* expression in cats that could only tolerate partial attenuation may be a response to an increased inflammatory associated with greater shunting. Decreased expression of hepatic SERPINB1 has been associated with tumour invasiveness and poor prognosis in humans with hepatocellular carcinoma [47]. The increased expression of both *HIF1a* and *NPR2* are consistent with an enhanced state of inflammation in cats that could only tolerate partial attenuation; both HIF1a and NPR2 have been shown to mediate the protective response to inflammation [48,49]. The exact relationship between enhanced *HIF1A*, *NPR2*, and *SERPINB1* gene expression and the inflammatory response in these cats remains to be established. 

In this study, we used a customised, multiplex XP-PCR assay, that enabled us to examine the expression of multiple hepatic transcripts in a single PCR reaction. Our previous investigations of hepatic gene expression in dogs with CPSS used qPCR [15]. It is possible that this specific methodology may have resulted in some of the discrepancies in our current results and those seen in dogs with CPSS. However, despite utilizing different chemistries, it is unlikely that these different methodologies are responsible for the observed gene expression changes in cat livers. We have previously analysed gene expression profiles from the same equine trophoblast RNA samples using multiplex XP-PCR and microarray analyses, and found them to be qualitatively identical [50]. A potential limitation of our current study is the discrepancy between the CPSS and control group populations, specifically with regard to age of animal. We cannot rule out that age-associated changes in hepatic gene expression might account for the differences between the CPSS and control cohorts; however, none of the genes investigated in the current study feature as age-dependent genes in livers from either rats or humans [51]. In addition to these age-associated issues, three of the control liver biopsies revealed some pathology (Table 2); although it is possible that this may have led to altered gene expression profiles, we did not observe any differences between hepatic gene expression in these cats compared to the other control cats with normal histopathology. As this was a pilot study using only small numbers of cats in each group corrections for performing multiple comparisons to minimise false discovery were not applied. Therefore, further studies are required to confirm these findings.

In summary, we described the first characterization of hepatic gene expression changes in liver biopsies from a small cohort of cats with CPSS. In contrast to the well-documented physiological down regulation of hepatic genes (e.g., *HGF*, *VEGFR2*, *CPS1*) reported in dogs with CPSS, our results indicated an *upregulation of* the expression of these markers of regeneration, angiogenesis, and urea formation. This suggested that the response of the liver in cats with CPSS may be governed by different factors to dogs. Further investigation is required to establish the physiological relevance of these altered adaptations in cats.

## 5. Conclusions

This study provides the first description of hepatic gene expression changes in a small cohort of cats with congenital portosystemic shunts. The increased expression of hepatic markers involved in liver regeneration, angiogenesis, urea cycle regulation and liver metabolism directly contrasts previously documented changes to hepatic gene expression in dogs with CPSS, suggesting that alternative mechanisms could regulate the hepatic response to portosystemic shunts in cats.

## Figures and Tables

**Figure 1 vetsci-11-00100-f001:**
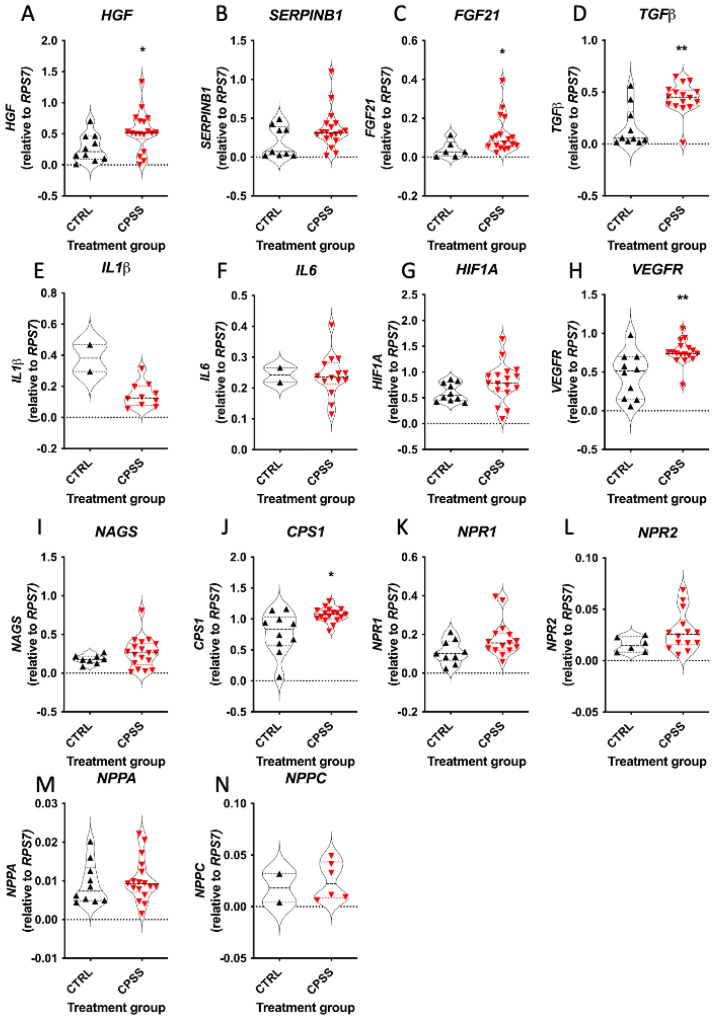
(**A**–**N**) Multiplex XP-PCR analysis of hepatic gene expression in cats with congenital portosystemic shunts (CPSS) compared with controls (CTRL). Data were normalised to the housekeeping gene (*RPS7*), and analysed by Mann-Whitney U test, with *p*-values represented as: * *p* < 0.05, ** *p* < 0.01) *n* = 18 for CPSS and *n* = 10 for control, although transcripts for every gene were not always detected in every sample).

**Figure 2 vetsci-11-00100-f002:**
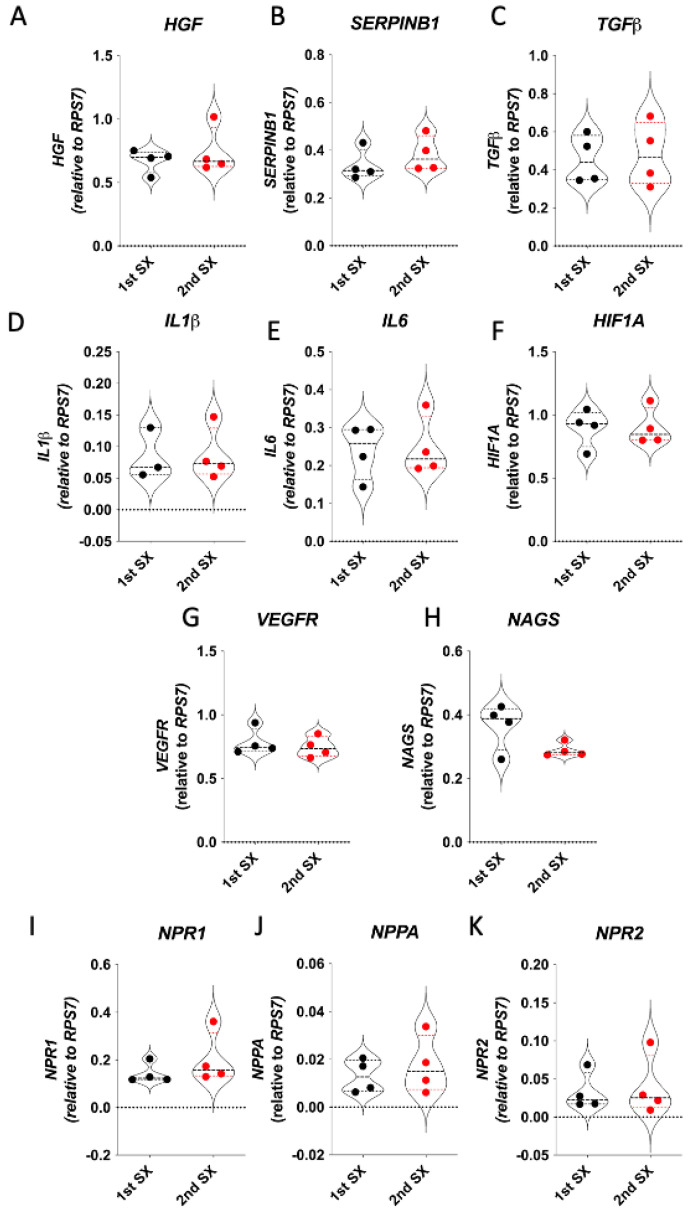
(**A**–**K**) Comparison of hepatic gene expression in cats with congenital portosystemic shunts at time of first and second surgery. Data were normalised to the housekeeping gene (*RPS7*) and analysed by Mann-Whitney U test. There were no significant differences observed; *n* = 4 cats.

**Table 1 vetsci-11-00100-t001:** Oligonucleotide primers used to detect hepatic gene expression.

Gene	Accession Number	Product Position (5′—3′) (bp)	Primer Sequences (5′—3′) (Forward and Reverse)	Product Size (including Universal Tag) (bp)
*HIF1A*	XM_003987716.1	576—675	AGGTGACACTATAGAATATCACTGCACAGGCCATATTC CGAACCACGACTAAACACTTAGGGATATCACTCAGCATG	137
*IL* *β*	NM_001077414	354—461	AGGTGACACTATAGAATAATCTGTGACACGTGGGATGA ACCACGACAGACCGAGTATGAGGGATATCACTCAGCATG	145
*IL6*	NM_001009211	446—575	AGGTGACACTATAGAATAAGATGCTGAAGCGTAAGGGA TTGTTAGTGGAGTGGGAAGCAGGGATATCACTCAGCATG	167
*SERPINB1*	XM_003985852	199—343	AGGTGACACTATAGAATAAGCATTGGCCATGATCTTTC TTGCGCCTCGGAGGATATAAAGGGATATCACTCAGCATG	182
*RPS7*	NM_001009832	100—222	AGGTGACACTATAGAATATGGAGATGAACTCCGACCTC AGGGCAAGGAGTTGACTTCAAGGGATATCACTCAGCATG	160
*TGF* *β*	XM_003997774	1537—1683	AGGTGACACTATAGAATAATCAACGGGTTCAGTTCCAG CTCGGGACCTGTGGTTGATGAGGGATATCACTCAGCATG	184
*VEGFR2*	XM_011281879	2739—2839	AGGTGACACTATAGAATAGCCTTATGATGCCAGCAAGT TCACTAGCTTCGTCTACGGAAGGGATATCACTCAGCATG	138
*CPS1*	XM_003991105	1306—1457	AGGTGACACTATAGAATACCACAATCACATCGGTTCTG GGCATTTTCGGTACTTCCTTAGGGATATCACTCAGCATG	189
*NPPA*	XM_003989498	217—324	AGGTGACACTATAGAATAGTCAGCTCTTGTGGCAAACA AAACCTCCTGTTCTACGGAAAGGGATATCACTCAGCATG	145
*NPPC*	XM_003991256	101—231	AGGTGACACTATAGAATATGCTCACGCTACTCTCGCT CAGTCTTCTTCCCGCTGTTTAGGGATATCACTCAGCATG	168
*NPR1*	XM_003999769	2366—2481	AGGTGACACTATAGAATATCCAGGATGGAGTCTAACGG GCTCCCCTGCATCTTTACTTAGGGATATCACTCAGCATG	153
*NPR2*	XM_003995612	1559—1696	AGGTGACACTATAGAATACTTTGACTTGGACGACCCAT TTCGACTACGACCTCTTCCTAGGGATATCACTCAGCATG	175
*FGF21*	XM_003997528	134—294	AGGTGACACTATAGAATATTCGACAGCGGTTCCTCTAC AAGTTTAGAACCCCCAGTTTAGGGATATCACTCAGCATG	198
*HGF*	NM_001009830	1835—2004	AGGTGACACTATAGAATACCTGCTGTCCTGGATGATTT TACCCCTTACTCTTTACGTCAGGGATATCACTCAGCATG	207
*NAGS*	XM_0112389272.1	805—992	AGGTGACACTATAGAATACTCTTCAGCAACAAGGGGTC AGGCAGATGCAGAGACTCCCAGGGATATCACTCAGCATG	225

**Table 2 vetsci-11-00100-t002:** Clinical characteristics of cats in congenital portosystemic shunt or control groups.

**Cat**	**Shunt Type**	**Age (days)**	**Sex**	**Breed**	**Attenuation**	**Interval between Surgeries (Days)**
1	Extrahepatic	180	ME	Maine Coon	Partial	
2	Extrahepatic	347	ME	DSH	Complete	
3	Extrahepatic	186	ME	DLH	Partial	
4	Intrahepatic	275	MN	DSH	Partial	
5	Extrahepatic	336	MN	Birman	Complete	
6	Extrahepatic	572	MN	DSH	Complete	
7	Extrahepatic	191	MN	DSH	Complete	
8	Extrahepatic	180	ME	DSH	Complete	
9	Extrahepatic	752	MN	DLH	Partial *	102
10	Extrahepatic	198	FE	DSH	Partial *	91
11	Intrahepatic	239	ME	BSH	Partial	
12	Extrahepatic	184	MN	DSH	Partial	
13	Extrahepatic	155	ME	Ragdoll	Partial	
14	Intrahepatic	307	FE	DSH	Partial *	110
15	Extrahepatic	284	FN	Ragdoll	Partial *	87
16	Extrahepatic	228	ME	DSH	Partial	
17	Extrahepatic	247	FN	Maine Coon	Partial	
18	Intrahepatic	1309	FN	Oriental	Partial	
**Control**	**Timing of Sample Collection**	**Age (days)**	**Sex**	**Breed**	**Liver** **Histopathology**	
1	Surgical	3061	FN	DSH	Liver mass (haematoma	
2	Post-mortem	2881	FN	DLH	Hepatocyte vacuolation	
3	Surgical	5353	FN	DLH	Biliary cysts	
4	Post-mortem	120	n/a	DSH	Normal	
5	Post-mortem	2191	MN	DSH	Normal	
6	Post-mortem	n/a	FE	DSH	Normal	
7	Post-mortem	2161	MN	DSH	Normal	
8	Post-mortem	550	MN	DLH	Normal	
9	Post-mortem	143	MN	Ragdoll x	Normal	
10	Post-mortem	303	MN	DSH	Normal	

Abbreviations. FE, Female entire; FN, female neutered; ME, male entire; MN, male neutered; BSH, British shorthair; DSH, domestic shorthair; DLH, domestic longhair; n/a, not available. * Denotes cats that had a repeat liver biopsy taken at second surgery.

**Table 3 vetsci-11-00100-t003:** Lack of relationship between hepatic gene expression in liver biopsies from cats with CPSS and plasma ammonia concentrations. Data shown are linear regression analyses comparing hepatic gene expression with blood ammonia concentrations. Abbreviation: nc, not calculated.

Gene	d.f.	*r*	*p*
*TGFb*	8	−0.03	0.92
*VEGFR2*	8	0.18	0.62
*IL1B*	5	0.48	0.27
*IL6*	8	−0.12	0.73
*NPR1*	7	−0.09	0.81
*NPR2*	8	0.36	0.35
*SERPINB1*	8	0.17	0.64
*NAGS*	8	0.14	0.71
*NPPA*	8	0.27	0.46
*NPPC*	nc	nc	nc
*HIF1A*	8	0.14	0.71
*HGF*	7	−0.08	0.84
*CPS1*	8	−0.47	0.17
*FGF21*	7	−0.41	0.27

## Data Availability

The data presented in this study are available on request from the corresponding author. The data are not publicly available due to patient privacy.

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
