# Peer review of "Hepatic Gene Expression of Angiogenic and Regeneration Markers in Cats with Congenital Portosystemic Shunts (CPSS)"

_vetsci, 2024, doi:10.3390/vetsci11030100_

Round 1
Reviewer 1 Report
Comments and Suggestions for Authors
1. How to diagnose CPSS, by CT or ultrasound, Please show the pictures for confirming the cpss .
Comments on the Quality of English LanguageQuality of English Language is good
Author Response
How to diagnose CPSS, by CT or ultrasound. Please show the pictures for confirming the CPSS.
All cats had their CPSS identified on ultrasound or CT scan prior to surgery. The shunt was confirmed on intra-operative mesenteric portovenography in all cats. We have added a sentence to confirm this, lines 87-89: “All cats had their CPSS diagnosed on ultrasound and / or computed tomography imaging prior to surgery.” Lines 100-101 confirm that portovenography was performed. As the imaging findings are not part of the study we are not sure of the value of adding pictures of the shunt imaging.
Reviewer 2 Report
Comments and Suggestions for Authors
This study describes the characterization of hepatic gene expression changes in liver biopsies from 18 cats with CPSS compared to that of 10 control cats . Authors also compare their data with those documented in dogs with portosystemic shunts.
The number of animals employed is reduced, but this is a pilot study and studies on hepatic gene expression has not previously been investigated in cats with CPSS. So, in my opinion the paper merits the acceptance in the present form.
I would like to congratulate Authors for the good-quality of their article
Author Response
Thank you for your supportive comments. We agree that due to the small sample size our conclusions are limited, until we can expand the scope of our studies. Nevertheless, our observations from this small cohort of cats with CPSS provide intriguing differences in the hepatic gene expression changes seen in dogs with CPSS compared to cats.
Reviewer 3 Report
Comments and Suggestions for Authors
Tivers and colleagues investigate the gene expression profile in cats affected by congenital portosystemic shunt and find a significant increase in TGFβ, VEGFR2, HGF, FGF21, and CPS1 in diseased animals.
The work is very interesting and well written, but the part of the histopathology results is missing. The histological characteristics of shunts should be described and shown with images. In addition, possible correlations between histological alterations (fibrosis, arteriolar proliferation and regeneration to mention a few) and the expression profile of pro-fibrogenic genes, associated with neo-angiogenesis or regeneration, should be investigated to have a more complete picture of the disease. .
Author Response
The histological characteristics of shunts should be described and shown with images. In addition, possible correlations between histological alterations (fibrosis, arteriolar proliferation and regeneration to mention a few) and the expression profile of pro-fibrogenic genes associated with neo-angiogenesis or regeneration, should be investigated to have a more complete picture of the disease.
This is a great suggestion and, hopefully, this pilot study will lead to further investigations of this nature. Histopathological changes in the liver from these cats were not assessed specifically as part of the study. The cats had routine histopathology performed as part of their clinical management and all were confirmed to have changes associated with congenital portosystemic shunting (portal vein hypoperfusion). We have previously published the histopathological findings of a cohort of cats with congenital portosystemic shunts: (Swinbourne F, Smith KC, Lipscomb VJ, Tivers MS. Histopathological findings in the livers of cats with a congenital portosystemic shunt before and after surgical attenuation. Veterinary Record 2013, 172: 362.) We therefore did not feel that it would be appropriate to repeat histopathological analysis in the current study as this was not the focus. In addition, with such a small number of cats and such a large number of variables, statistical analysis of histopathological changes with gene expression would have been unreliable with significant risk of type I and type II statistical errors. We have added the following to lines 143-145:
“Histopathological findings were consistent with portal vein hypoperfusion in all cats with CPSS, as previously described (Swinbourne 2013).”
Round 2
Reviewer 3 Report
Comments and Suggestions for Authors
The authors responded to my comment. The work can therefore be accepted in its current form.
Author Response
Many thanks for your suggestions, and your efforts in reviewing our manuscript.